# Anti-Inflammatory Activity of Mulberry Leaf Flavonoids In Vitro and In Vivo

**DOI:** 10.3390/ijms23147694

**Published:** 2022-07-12

**Authors:** Ziwei Lin, Tiantian Gan, Yanzhen Huang, Lijun Bao, Shuang Liu, Xiaopeng Cui, Hexin Wang, Feng Jiao, Minjuan Zhang, Chao Su, Yonghua Qian

**Affiliations:** The Sericultural and Silk Research Institute, College of Animal Science and Technology, Northwest A&F University, Xianyang 712100, China; linxiaoliu510@163.com (Z.L.); gantiantian1987@163.com (T.G.); whyiggy@126.com (Y.H.); baolijun@nwafu.edu.cn (L.B.); 1732546764@163.com (S.L.); cuixpssyjs@163.com (X.C.); wanghexin1111@163.com (H.W.); fjiao@nwsuaf.edu.cn (F.J.); mjzhang1008@nwsuaf.edu.cn (M.Z.)

**Keywords:** mulberry leaf, flavonoids, anti-inflammatory, LC-MS, ulcerative colitis

## Abstract

Mulberry (*Morus alba* L.) is a flowering tree traditionally used in Chinese herbal medicine. Mulberry leaf flavonoids (MLFs) have been reported to exert important anti-inflammatory and antioxidant properties. The purpose of this study was to select the MLF with the best anti-inflammatory and antioxidative activities from MLFs eluted by different ethanol concentrations (30%, 50%, and 75%) and explore its pharmacological properties. Three types of MLFs inhibited the production of nitric oxide (NO), prostaglandin E2 (PGE2), inducible nitric oxide synthase (iNOS), cyclooxygenase-2 (COX-2), and inflammatory cytokines in lipopolysaccharide (LPS)-induced RAW 264.7 cells. All MLFs boosted the antioxidative capacity by decreasing the reactive oxygen species (ROS) production and the scavenging of 2,2-diphenyl-1-picrylhydrazyl (DPPH) free radicals and improving the metal ion chelating activity and reducing power. The results revealed that the MLFs eluted by 30% ethanol exhibited the best anti-inflammatory and antioxidative activities. A nontargeted metabolomic analysis was used to analyze 24 types of differential flavonoids between the MLFs. Quercetin, kaempferol, and their derivatives in 30%MLF were more abundant than the other two MLFs. Furthermore, we evaluated the pharmacological activities of 30%MLF in dextran sodium sulfate (DSS)-induced ulcerative colitis (UC) mice. The 30%MLF could alleviate the clinical symptoms, reduce the secretion of inflammatory cytokines, and inhibit the activation of the inflammatory pathway in DSS-induced colitis mice. This study will provide valuable information for the development of MLFs eluted by 30% ethanol as a functional food.

## 1. Introduction

Inflammation is a basic pathological process involved in defensive responses triggered by internal or external stimuli [1]. The main symptoms of inflammation are calor (heat), rubor (redness), tumor (swelling), dolor (pain), and loss of function [2]. Moderate inflammatory responses have positive effects that are helpful and necessary for the body to resist harmful stimuli [3]. However, under certain conditions, such as excessive and chronic inflammation, they may be involved in a range of acute and chronic diseases, including inflammatory bowel disease (IBD), cardiovascular disease (CVD), type 2 diabetes mellitus (T2DM), rheumatoid arthritis (RA), and atherosclerosis (AS) [1,2]. There is a close link between inflammation and oxidative stress, wherein the involvement of oxidative stress is a common underlying factor in the pathogenesis of most chronic inflammatory diseases [4]. Oxidative stress is the imbalance between cellular oxidants and antioxidants, which leads to the poor abolition of reactive oxygen and nitrogen species (ROS and RNS) formed in the cell [5]. Under normal physiological conditions, oxidants and antioxidants in the body maintain a redox balance. When the balance is disrupted, oxidative stress occurs and damages the proteins, lipids, and DNA in the cell.

Genetic and environmental factors, such as diet, figure into inflammation and antioxidation [6]. Recently, a healthy diet that includes vegetables and fruits has attracted attention for its anti-inflammatory and antioxidant effects [7]. Scientific evidence has demonstrated that the health benefits of eating vegetables and fruit are a result of the combined action of numerous bioactive constituents, including carotenoids, minerals, polysaccharide, flavonoids, and vitamins [8,9]. Flavonoids, a class of plant-derived dietary compounds, are secondary metabolites abundantly found in fruits, vegetables, and herbal medicine, among other sources [10,11]. Previous studies have reported that flavonoids exhibit anti-inflammatory and antioxidative activities in other plants [12,13]. Mulberry (*Morus alba* L.), a fast-growing tree, is found in warm and humid climates and can withstand cold and drought. The different parts of mulberry are rich in flavonoids and exert anti-inflammatory and antioxidative activities, including the root bark, fruits, and leaves [14,15]. Mulberry leaves are a food source for silkworms and used for medicine, human food, and animal husbandry, which have been attributed to its abundant active ingredients and nutritional value [16]. Their leaves and leaf-derived extracts are also rich in flavonoids that possess anti-inflammatory and antioxidative activities [17,18,19]. However, which flavonoid types have the best anti-inflammatory and antioxidant activities remains to be determined. Owing to the wide variety of mulberry species, the composition and efficacy of mulberry leaf flavonoids (MLFs) may vary across varieties. Thus, it is necessary to further study its components and pharmacological properties [16].

Ulcerative colitis (UC) is often collectively referred to as IBD, together with Crohn’s disease (CD), and its incidence rate has increased significantly in the past two decades, placing a heavy burden on health systems in various countries [20]. The typical clinical symptoms of UC include abdominal pain, weight loss, diarrhea, and bloody stools [21], which seriously affect the health and daily life of patients. Currently, the most common drugs used to treat UC are 5-aminosalicylic acid drugs, corticosteroids, steroids, and immunosuppressants, which are very expensive. Although these medications provide some benefits to patients, they also have drawbacks. The long-term administration of these medications may result in a weakened immunity, making patients more susceptible to infections [22]. Consequently, screening for low-cost, low-toxicity, and effective candidate therapies from natural plant-derived bioactive substances is important. Recently, various studies have reported the intestinal anti-inflammatory activity of natural products, particularly flavonoids [23].

Therefore, the aim of this study was to select the MLF with the best anti-inflammatory and antioxidative activity and evaluate its pharmacological properties in dextran sodium sulfate (DSS)-induced UC mice. We extracted three MLFs from a fruit mulberry variety, which were then eluted with 30%, 50%, and 75% ethanol, and compared the anti-inflammatory and antioxidative activities of the three MLFs in lipopolysaccharide (LPS)-induced RAW 264.7 cells. Their antioxidative properties were tested in vitro. The MLF eluted with 30% ethanol had the highest total flavonoid content (TFC) and showed the best anti-inflammatory and antioxidant activities. Additionally, we preliminarily determined that quercetin, kaempferol, and their derivatives may be the main pivotal bioactive flavonoids. More importantly, we explored the pharmacological properties of the MLF eluted with 30% ethanol in DSS-induced UC mice. Physiological changes, the secretion of inflammatory cytokines, and activation of the inflammatory pathway were analyzed in order to explore its protective mechanism.

## 2. Results and Discussion

### 2.1. Cytotoxicity of the MLFs in RAW264.7 Cells

LPS-induced RAW 264.7 cells are the most commonly used models to assess the potential bioactivity and anti-inflammatory properties of natural products [24,25]. A CCK-8 assay was performed to estimate the cell viability of RAW 264.7 cells treated with the MLFs and LPS to avoid cell death caused by excess concentrations. The 5 μg/mL, 50 μg/mL, 150 μg/mL, and 250 μg/mL doses of MLFs contained 0.0025%, 0.025%, 0.075%, and 0.125% dimethyl sulfoxide (DMSO), respectively. Thus, we also tested the cytotoxicity of different concentration of DMSO. In Figure 1D, the 0.125% DMSO exhibited significant cytotoxicity toward RAW 264.7 cells after 24 h and 48 h (*p* < 0.05). Moreover, as shown in Figure 1A,C, 30%MLF and 75%MLF at 250 μg/mL decreased the cell viability after 48 h (*p* < 0.05). These results demonstrated that 250 μg/mL of MLFs could not be used in further experiments. As shown in Figure 1A–D, 150 μg/mL of MLFs and 0.075% DMSO had no effect on cell viability after 12, 24, and 48 h (*p* > 0.05). Therefore, MLFs of 150 μg/mL were selected for use in further experiments. Interestingly, the MLFs showed an increase in the viability of the RAW 264.7 cells, maybe due to some immunostimulatory components in the extract from the medicinal plant [26,27]. LPS, an endotoxin, is a major component of the Gram-negative bacterial cell wall [4]. It is one of the most extensively used effective stimulators to activate macrophages and trigger an inflammatory response [28]. Additionally, the cytotoxicity of LPS was also investigated. As shown in Figure 1E, 1 μg/mL of LPS treatment had no cytotoxicity on the cells (*p* > 0.05).

### 2.2. MLFs Decreased Nitric Oxide (NO) and Prostaglandin E2 (PGE2) Production in LPS-Induced RAW 264.7 Cells

The pathogenesis of inflammation involves inflammatory responses mediated by NO and PGE2. In this process, the expression of inducible nitric oxide synthase (iNOS) and cyclooxygenase-2 (COX-2) genes increase significantly, which thereby promotes NO and PGE2 synthesis [29]. Therefore, regulating the overexpression of these mediators will decrease the damage caused by inflammatory responses. NO is an important indicator of inflammation that participates in various molecular and biological pathways and plays a key role in many physiological and pathophysiological processes [30]. To explore the anti-inflammatory activities of MLFs, we estimated the NO scavenging ability of the MLFs (Figure 2A). Compared to the control, the NO production was almost six times greater in the LPS-treated groups (*p* < 0.01). NO is beneficial in neurological functions and defense mechanisms; however, excessive NO production has been associated with various disease complications, especially inflammatory diseases [31]. In this study, MLF pretreatments led to a significant decrease in NO release compared to the LPS group (*p* < 0.01), while 30%MLF showed better effects than 50% and 75%MLF (*p* < 0.05). iNOS can induce a large amount of NO within a short amount of time [32], thus mediating the occurrence and development of inflammatory diseases [33]. Therefore, inhibiting iNOS expression is conducive to reducing NO production. The MLFs significantly decreased the LPS-induced iNOS mRNA expression in RAW 264.7 cells (*p* < 0.05), and the 30%MLF treatment showed a better inhibition of iNOS mRNA expression than the 50%MLF treatment (*p* > 0.05) and 75%MLF treatment (*p* < 0.05) (Figure 2B). Compared with the control group, the LPS treatment also led to an obvious increase in the PGE2 content and COX-2 expression in RAW 264.7 cells (*p* < 0.01) (Figure 2C,D). PGE2 is another inflammatory mediator and a prostaglandin mainly produced by COX-2, both of which are involved in a series of physiological and pathological processes [34]. Compared to the LPS group, 30%MLF markedly lowered PGE2 (*p* < 0.01) and COX-2 production (*p* < 0.05), which are closely associated with oxidative stress (Figure 2C,D), while the 50% and 75%MLF treatments did not significantly decrease the LPS-induced COX-2 expression (*p* > 0.05) (Figure 2D). These results indicated that 30%MLF showed the best inhibition of NO, iNOS, PGE2, and COX-2 production among the three types of MLFs and relieved the anti-inflammatory reactions by diminishing iNOS and COX-2 to decrease the levels of NO and PGE2. Previous studies have reported that phenolic compounds contributed to the treatment of chronic inflammatory diseases related to excessive NO production and are important for restraining PGE2 in inflammatory tissues [35,36]. These results indicated that 30%MLF has the therapeutic potential to prevent or treat inflammatory diseases.

### 2.3. MLFs Inhibited Inflammatory Cytokine Secretion in LPS-Induced RAW 264.7 Cells

Inflammatory cytokines are involved in inflammatory responses that directly damage the vascular endothelium and lead to increased vascular permeability [37]. The mechanism that controls inflammation suppresses inflammatory cytokines and mediates their production and/or function. To verify the anti-inflammatory effects of the MLFs, we examined the expression of various inflammatory cytokines by using real-time quantitative polymerase chain reaction (RT-qPCR) and enzyme linked immunosorbent assay (ELISA).

The results revealed that LPS induced NO production. In turn, NO can induce tumor necrosis factor-α (TNF-α) production, which is a proinflammatory cytokine [38]. LPS induction considerably increased TNF-α, interleukin-1β (IL-1β), interleukin-6 (IL-6), and monocyte chemoattractant protein-1 (MCP-1) expression when compared to the control (*p* < 0.01) (Figure 3). Compared to the LPS group, 30% and 75%MLF significantly inhibited TNF-α production, especially 30%MLF (*p* < 0.01) (Figure 3A,E). TNF-α can activate a cytokine cascade in inflammatory responses and stimulate IL-1β, IL-6, and MCP-1 expression [39]. IL-1β is important for the initiation of inflammatory immune responses and may be a promising therapeutic target for inflammatory diseases [40]. The MLFs greatly decreased the IL-1β secretion in LPS-induced RAW 264.7 cells, especially 30%MLF (*p* < 0.01) (Figure 3B,F). IL-6 is a multifunctional cytokine produced during inflammation [35]. It can stimulate the production of most acute-phase proteins in inflammatory responses and plays a critical role in the host defense against invasive infections, thereby directly reflecting the existence and intensity of inflammation [41]. In this study, 30% and 75%MLF significantly reduced LPS-induced IL-6 expression (*p* < 0.01) (Figure 3C,G). As shown in Figure 3F,G, 30%MLF showed a better inhibition of IL-1β and IL-6 production than 50% and 70%MLF (*p* < 0.05). The MLFs significantly downregulated LPS-induced MCP-1 expression, especially 30%MLF (*p* < 0.01) (Figure 3D), which recruits immune cells to inflammation sites to regulate the tissue injury, repair, and regeneration [42]. Macrophages stimulated by LPS result in the release of proinflammatory cytokines (TNF-α, IL-6, and IL-1β); mediators (NO and PGE2); and enzymes (iNOS and COX-2) [43]. Therefore, inhibiting inflammatory cytokines, mediators, and enzymes is a promising strategy for treating inflammatory diseases. In this study, 30%MLF effectively inhibited TNF-α, IL-1β, IL-6, and MCP-1 expression in LPS-induced RAW 264.7 cells. These results indicated that 30%MLF may show better anti-inflammatory activities than the other two MLFs.

### 2.4. MLFs Inhibited ROS Production in LPS-Induced RAW 264.7 Cells

Inflammation responses are closely related to oxidative stress. For example, the increased expression of inflammatory factors is often accompanied by excessive mitochondrial ROS production [44]. The transmission of inflammatory signals in macrophages is mediated by ROS, signaling molecules closely related to the host defense responses, gene transcription, and apoptosis [45], which results in the hyperactivation of inflammatory responses, leading to tissue damage and excessive ROS production, and related species may cause oxidative stress phenomena [46]. Oxidative stress and inflammatory responses are important factors in several human diseases, which lead to a vicious cycle [47,48]. Therefore, we assessed the level of ROS in LPS-induced RAW 264.7 cells. Compared to the blank control, the LPS treatment significantly increased the ROS level in RAW 264.7 cells (Figure 4). However, less fluorescence was observed in the cells treated with the MLFs, indicating that the ROS level was inhibited. Previous research has shown that an MLE (40% ethanol and hydroalcohol extracts mixed together) exhibited antioxidant potential by inhibiting resistin-induced ROS production, fractaline, and P-selectin expression in human endothelial cells [49]. In this study, the MLFs strongly inhibited the total ROS and acted as potential antioxidants.

### 2.5. Antioxidant Activities of the MLFs

To evaluate the antioxidant activities of the MLFs, the DPPH radical scavenging activity, metal ion chelating activity, and reducing power were determined. Free radicals, together with other oxygen derivatives, are byproducts of biological redox reactions, which are related to certain diseases, such as diabetes, liver cirrhosis, nephrotoxicity, and cancer [50]. DPPH is a stable free radical that is widely used to evaluate the antioxidant activity of different compounds [51]. In this study, 30% (12.57 ± 0.90 μg/mL) and 50%MLF (12.35 ± 0.55 μg/mL) exhibited a lower IC_50_ for DPPH radical scavenging activity, but no significant difference between 30% and 50%MLF was detected (Table 1). Compounds with a lower IC_50_ have a higher DPPH scavenging activity. It was reported that flavonoids are the main components of most plants and have antioxidant and free radical scavenging activities [52]. Our results indicated that 30% and 50%MLF were powerful free radical scavengers. Additionally, ferrous iron (Fe^2+^), the most effective oxidant associated with food systems, can be removed by chelating agents, which prevent oxidative stress-induced diseases [53]. Therefore, this experiment tested the ability of the MLFs to chelate Fe^2+^ by competing with another chelating agent, potassium ferricyanide. The results revealed that 30%MLF had the lowest IC_50_ (310.56 ± 9.72 μg/mL) for chelating activity among the three types of MLFs (*p* < 0.05) (Table 1). Previous studies also proved that the antioxidant capacity is related to the reducing capacity [51,54,55]. In this study, the three MLFs had a certain degree of reducing power and potential antioxidant activity. Moreover, the reducing power of 30%MLF (41.69%) was higher than 50% (38.95%) (*p* > 0.05) and 75%MLF (31.11%) (*p* < 0.05) (Table 1). Therefore, we concluded that the MLFs exhibited free radical scavenging effects on DPPH, had a stronger Fe^2+^-chelating ability, and had better reducing power assays. Collectively, these results indicated that the MLFs are primary antioxidants.

### 2.6. TFC of the MLFs

We tested and compared the total content of the three types of MLFs. The results revealed that 30%MLF had the highest TFC purity (72.89%), while 75%MLF had the lowest (55.12%) (Table 2). Owing to the varying polarities of flavonoids, the selection of the extraction solvent is a critical step for extracting the maximum quantity of active constituents [56]. Our results indicated that 30% ethanol may be a better concentration for eluting MLFs with a higher flavonoid content. The types of MLFs obtained in this study are shown in Appendix A, which are consistent with previously reported constituents in MLFs [17,18].

### 2.7. Differential Flavonoids between the MLFs

The studies described above showed that 30%MLF exerted better antioxidant and anti-inflammatory effects. Therefore, we tested the differential flavonoids between the MLFs. A nontargeted metabolomic analysis was performed to detect the differences in flavonoids among the MLFs. After the peak alignment and the removal of the even *m/z* ions, a total of 1164 ions were obtained. Through a principal component analysis (PCA) of 1164 ions, we found that the MLFs eluted with 30%, 50%, and 75% ethanol divided into three groups (Figure 5A), indicating that the types of flavonoids eluted with different ethanol concentrations were significantly different. The differences in flavonoids among the MLFs were further investigated by a partial least-squares discriminant analysis (PLS-DA). The results revealed that the samples also divided into three groups (Figure 5B); thus, the PLS-DA model was reliable by cross-validation (Figure 5C). The differential compounds are shown in Figure 5D. A total of 237 ions (variable importance in the projection, VIP > 1) were screened as the differential compounds, and finally, 24 differential flavonoids were identified by comparing the accurate mass and tandem MS spectra with the HMDB database, which mainly consisted of quercetin and kaempferol glycosides (Table 3). Among them, four flavonoids were identified based on the authentic standards.

Chemically, flavonoids are comprised of two aromatic rings, A and B, which are linked by a heterocyclic ring, identified as ring C [57,58]. Flavonoids have been classified into different subclasses based on the oxidation and degree of unsaturation of ring C, as well as the connection position of ring B [59]. The subclasses of flavonoids include flavonols, flavones, flavanols, flavanones, isoflavones, and anthocyanidins. A previous investigation indicated that flavonols are the most abundant flavonoids found in food [60]. MLFs contain a large amount of flavonol derivatives, mainly glycosylated forms of quercetin and kaempferol, which have high antioxidative activities [61,62]. We found that the contents of quercetin and kaempferol glycosides in 30%MLF were greater than 50% and 75%MLF (Figure 6). The solvent selection is important for flavonoid extraction, which determines the quantitative and qualitative components of these compounds to a certain extent; the TFC and antioxidant capacities in the same plant may widely vary depending on the extraction method and conditions [63]. Our results indicated that quercetin, kaempferol, and their derivatives in 30%MLF may be the main flavonoids involved in anti-inflammatory and antioxidative activities when compared to 50% and 75%MLF. Moreover, using 30% ethanol to elute MLFs may more effectively obtain compounds that have anti-inflammatory and antioxidative capacities.

At the molecular level, kaempferol regulates many key elements in the cell signal transduction pathways related to inflammation [63]. A previous study found that a diet high in flavonols (especially kaempferol) was associated with decreased levels of the inflammatory cytokine IL-6 [64]. Quercetin is one of the most abundant flavonoids found in nature, which usually exists in the form of glycoside and displays inflammatory and antioxidant effects [65]. Mulberry leaves have a high quercetin content, which can reduce the oxidation process in vivo and in vitro [66]. Quercetin 3-(6malonylglucoside) is the most important flavonoid with antioxidant potential found in mulberry leaves [18,67]. It has been reported that kaempferol and quercetin inhibit iNOS mRNA and protein expression, as well as NO production in LPS-induced J774 and RAW 264.7 cells, which thereby reduce the inflammatory response [68,69]. Additionally, a separate study found that kaempferol treatment can relieve diabetic neuropathic pain induced by the intraperitoneal injection of streptozotocin in Swiss mice by decreasing the inflammatory mediators (NO, IL-1β, and TNF-α) and oxidative stress (GSH and MDA production) [70]. *M. nigra* leaves have higher phenolic compound contents and antioxidant activity than the pulps, which are measured by neutralizing the DPPH radicals [71]. A previous study showed that nine flavonoids, including kaempferol, quercetin, and their derivatives, isolated from mulberry leaves exhibited significant radical scavenging effects [72]. The compounds identified in 30%MLF in this study may possess the main anti-inflammatory and antioxidative capacities, which is similar to the results of previous studies.

### 2.8. Treatment with 30%MLF Alleviated the Symptoms of DSS-Induced UC in Mice

The studies described above indicated that 30%MLF exhibited better anti-inflammatory and antioxidant activities and higher TFC than 50% and 75%MLF. Therefore, we explored the pharmacological properties of 30%MLF in DSS-induced colitis mice.

Figure 7 shows the body weight change, food intake, disease activity index (DAI), colon length, and spleen index of mice. As expected, mice treated with DSS presented symptoms such as the loss of body weight, decreased food intake, diarrhea, and bloody stools, indicating that the colitis model was successfully induced by 3% DSS.

Body weight loss is related to disease severity in DSS-induced colitis model mice [73]. As shown in Figure 7A, the body weight of the DSS group began to decrease on the fourth day after being induced by DSS. By the end of the experiment, DSS supplementation resulted in a body weight decrease of nearly 25% compared to the control group (*p* < 0.05). The food intake of mice in the DSS group also declined compared to the control group (*p* < 0.05, Figure 7B), while the 30%MLF treatment (30%MLF + DSS group) significantly reduced these trends (*p* < 0.05, Figure 7A,B).

The DAI scores demonstrated the development of colitis [74]. Figure 7C displays the DAI scores of the different groups. As expected, the DAI scores of the DSS group increased gradually and were much higher compared to the control group during the DSS intervention (*p* < 0.05). In addition, the ingestion of 30%MLF (the 30%MLF + DSS group) delayed the elevation of the DAI compared to the DSS group (*p* < 0.05). These results indicated that 30%MLF supplementation could alleviate the loss of body weight, diarrhea, and the severity of fecal blood of DSS-induced UC in mice.

Colon shortening is another typical indicator of colitis severity [75]. As shown in Figure 7D,E, the colon lengths of the DSS group were significantly shorter compared to the control group. In contrast, the 30%MLF treatment (30%MLF + DSS group) significantly prevented colon shortening (*p* < 0.05). In addition, the spleen is one of the major immune organs of human body, and spleen hypertrophy is related to DSS-induced colitis [76]. Therefore, the spleen index was also tested. As shown in Figure 7F, the administration of 30%MLF (30%MLF + DSS group) also significantly alleviated DSS-induced spleen edema (*p* < 0.05).

The above results indicated that the 30%MLF treatment could mitigate these symptoms of DSS-induce colitis in mice.

### 2.9. Treatment with 30%MLF Relieved the Morphological Damage Caused by DSS-Induced UC in Mice

The histopathological injuries of mice colon tissues from different groups were detected using hematoxylin and eosin (H&E). As shown in Figure 8A, the colon tissues from the DSS group had the typical pathological characteristics of colitis, including the infiltration of inflammatory cells, damaged glands, the loss of crypts, and colonic mucosa erosion. In addition, the histological scores of these mice were nearly 12 times greater than those of the control group (*p* < 0.05, Figure 8B). These changes in the colon tissues obtained from the DSS group mice further demonstrated that 3% DSS treatments could successfully establish a mice colitis model. Under DSS administration accompanied by 30%MLF treatment, the colon tissues exhibited relatively intact intestinal mucosa and glands, low infiltration of the inflammatory cells, and little crypt damage. The histological scores of the 30%MLF + DSS group were significantly lower than those of the DSS group (*p* < 0.05, Figure 8B).

Dodda et al. [77] reported that quercetin might show good potential in suppressing acetic acid-induced IBD by ameliorating the alteration of the colon morphological parameters. Hong et al. [78] reported that supplementation with quercetin aglycone and quercetin aglycone with monoglycosides, even at a low dose, could alleviate the histological status (scattered villi and neutrophil infiltration) in DSS-induced colitis. The results shown in Figure 8 indicate that the 30%MLF treatment had a protective effect on the colon injury induced by DSS, and the mechanism of this effect might be associated with anti-inflammatory activity.

### 2.10. Treatment with 30%MLF Reduced Inflammatory Cytokine Secretion of DSS-Induced UC in Mice

Inflammation is a major feature of colitis, and the proinflammatory cytokines (TNF-α, IL-1β, and IL-6) are involved in the development and pathogenesis of UC [79]. The increased secretion of proinflammatory cytokines causes colon damage during colitis [80]. Therefore, inhibiting these proinflammatory cytokines could mitigate UC in mice. In this study, the content of inflammatory cytokines TNF-α, IL-1β, and IL-6 in colon tissues was analyzed using the corresponding ELISA kit. Compared to the mice in the control group, there was a significant increase in TNF-α and IL-1β secretion in the DSS group (*p* < 0.05, Figure 9A,B). This effect might aggravate the inflammatory response in the intestine [76]. Importantly, TNF-α and IL-1β can promote neutrophil infiltration or increase the intestinal permeability. These effects led to diarrhea. The DSS also increased IL-6 secretion (*p* < 0.05, Figure 9C), which may have activated Th17 cells to secrete inflammatory cytokines and cause colitis in mice [76].

However, TNF-α, IL-1β, and IL-6 secretion was remarkably inhibited in the 30%MLF + DSS group compared with the DSS group (*p* < 0.05, Figure 9). Previous studies reported that flavonoids might mitigate colitis through prohibiting the secretion of inflammatory cytokines. For example, a naringenin pretreatment could decrease TNF-α, IL-1β, and IL-6 in acetic acid-induced UC in mice [81]. Troxerutin, a kind of flavonoid, can alleviate DSS-induced colitis in mice by inhibiting TNF-α, IL-1β, and IL-6 in the colon tissue [82]. Lin et al. [83] reported that dietary quercetin could suppress the production of proinflammatory cytokines (TNF-α and IL-6) in Citrobacter rodentium-induced colitis mice. Therefore, the results in this study may indicate that 30%MLF alleviated the inflammatory response by inhibiting the levels of the proinflammatory mediators and finally regulating DSS-induced colitis in mice.

### 2.11. Treatment with 30%MLF Inhibited the Toll-like Receptor 4 (TLR4)/Myeloid Differentiation Factor 88 (MyD88) Pathway Activation of DSS-Induced UC in Mice

The release of proinflammatory cytokines will be caused by the activation of the TLR4/MyD88 signaling pathway [84]. TLR4/MyD88, a classic inflammatory pathway, is activated in mice with IBD [85,86]. Therefore, the expression of TLR4 and MyD88 were detected by an immunohistochemical analysis in the present study. TLR4 is a key transmembrane protein that mediates the immune response by recognizing molecules derived from pathogens and endogenous host-derived molecules [85]. MyD88, a central regulator of innate immunity, is the downstream regulator of the Toll-like receptors (TLRs) [87]. As expected, the DSS treatment increased the TLR4 and MyD88 expression compared to the control group (*p* < 0.05, Figure 10). However, the TLR4 and MyD88 expression levels of the 30%MLF + DSS group mice were significantly decreased in comparison with those of the DSS group (*p* < 0.05). A previous study reported that the detection of TLR4 and MyD88 in intestinal mesenchymal cells (IMCs) led to a significant reduction of intestinal tumors [87]. Therefore, 30%MLF may alleviate DSS-induced colitis through inhibiting the activation of the TLR4/MyD88 pathways.

## 3. Materials and Methods

### 3.1. MLF Extraction

Hongguo2 (*Morus atropurpurea*), a widely cultivated mulberry fruit tree in Northwest China, was bred by the Institute of Sericulture and Silk, Northwest A&F University. The leaves of Hongguo2 used in this study were authenticated and supplied by this institute. The extraction process followed previously described methods with minor modifications [24,88]. Fresh mulberry leaves without disease or insect pests were picked from plants in the same leaf position. Briefly, the dried and powdered whole plant (500 g) samples were refluxed with 85% ethanol three times. For the first extraction, the sample had a solvent ratio of 1:15 for 2.5 h. The second extraction was performed with a solid-to-liquid ratio of 1:10 for 2 h. The third extraction had a solid-to-liquid ratio of 1:8 for 1.5 h. The three drug solutions were combined and concentrated under reduced pressure, then left overnight and filtered. Next, the filtrate passed through processed AB-8 macroporous adsorption resin (Sunresin, Xi’an, China) and was washed with water. Then, the sample was eluted in a 30%, 50%, and 75% ethanol sequence to obtain three drug solutions. The solutions were vacuum-concentrated to remove the ethanol and freeze-dried to obtain three MLFs, named 30%MLF (4.29 g), 50%MLF (4.00 g), and 75%MLF (5.29 g). All extracts were stored at −20 °C for further analysis.

### 3.2. Cell Cultures

RAW 264.7 cells of a mouse macrophage cell line were purchased from the National Collection of Authenticated Cell Cultures (Shanghai, China). These are important inflammatory cells that play a critical role in the initiation and process of inflammatory responses. RAW 264.7 cells were cultured in high-glucose Dulbecco’s modified Eagle’s medium (DMEM) (Gibco, Grand Island, NY, USA) supplemented with 10% FBS (Gibco, Grand Island, NY, USA), 100 U/mL penicillin, and 100 μg/mL streptomycin (Gibco, Grand Island, NY, USA) in a humidified incubator containing 5% CO_2_ at 37 °C.

### 3.3. Cell Viability Assay

The analytical method was modified according to previous studies [89]. The effects of the samples on the viability of RAW 264.7 macrophages were measured using the CCK-8 Cell Proliferation and Cytotoxicity Assay Kit (CA1210; Beijing Solarbio Science & Technology Co., Ltd., Beijing, China). After overnight culturing in a 96-well plate (2 × 10^4^ cells/well), the cells were treated with various MLF concentrations (0, 5, 50, 150, and 250 μg/mL), which were dissolved in DMSO (Sigma-Aldrich, St. Louis, MO, USA) for 12, 24, and 48 h. According to the manufacturer’s instructions, a 10 μL CCK-8 solution was added to each well and incubated at 37 °C for 3 h. The absorbance was measured by a microplate reader (Thermo Fisher Scientific, Waltham, MA, USA) at 450 nm. Each sample had three replicates.

### 3.4. NO Production Assay

The analytical method was modified according to previous reports [89]. NO production was determined using a commercial NO assay kit (S0021S; Beyotime Institute of Biotechnology, Shanghai, China). Briefly, RAW 264.7 macrophages (1 × 10^5^ cells/well) were plated onto 24-well plates and pretreated with MLFs (the concentration was based on the cell viability assay results) for 12 h prior to stimulation with 1 μg/mL LPS (*Escherichia coli* O111:B4; Sigma-Aldrich, St. Louis, MO, USA) for 12 h. The culture supernatant from each well was collected and used to measure the NO production. Following the manufacturer’s instructions, 50 µL of the supernatant were mixed with an equal volume of Griess reagents I and II and reacted in a 96-well plate at room temperature for 10 min. The absorbance was measured at 540 nm on a microplate reader. Each sample had three replicates.

### 3.5. ROS Level Detection

The analytical method was modified from previously described methods [89]. The intracellular ROS level was determined using a ROS Assay Kit (S0033S; Beyotime Institute of Biotechnology, Shanghai, China). RAW 264.7 macrophages (5 × 10^5^ cells/well) were plated onto 6-well plates and pretreated with MLFs (the concentration was based on the cell viability assay results) for 12 h prior to stimulation with 1 μg/mL LPS for 12 h. Briefly, the cell culture medium was replaced with 10 μmol/L DCFH-DA and incubated at 37 °C for 20 min in the dark. Then, the extracellular DCFH-DA was removed and washed three times with a serum-free medium. Microscopic photographs were obtained in a random way using a digital camera.

### 3.6. ELISA Test

The analytical method was modified from previously described methods [44]. The secretion levels of PGE2, TNF-α, IL-1β, and IL-6 in LPS-induced RAW 264.7 cells were measured using an ELISA kit (Fankewei, Shanghai, China). After overnight culturing in a 24-well plate (1 × 10^5^ cells/well), RAW 264.7 macrophages cells were pretreated with MLFs (the concentration was based on the cell viability assay results) for 12 h. Then, 1 μg/mL LPS was added, and the reaction proceeded for another 12 h. The supernatants were collected from the medium after centrifugation and used to measure the PGE2, TNF-α, IL-1β, and IL-6 concentrations following the manufacturer’s instructions. The colon tissues of mice were homogenized with phosphate-buffered saline (PBS, Servicebio Technology Co., Ltd., Wuhan, China) under low temperatures and then centrifuged at 5000× *g* for 5 min to collect the supernatant. The levels of TNF-α, IL-1β, and IL-6 were tested using the ELISA assay following the manufacturer’s instructions. The ELISA kits were purchased from MultiSciences (MultiSciences Biotech Co., Ltd., Hangzhou, China). Each sample had three replicates.

### 3.7. RNA Extraction and RT-qPCR

The total RNA of the RAW 264.7 cells was extracted using a Trizol reagent (Invitrogen, Carlsbad, CA, USA) following the manufacturer’s instructions. The RNA purity was determined with an absorption ratio (A_260nm_/A_280nm_) between 1.8 and 2.0. The RNA was reverse-transcribed into cDNA using a commercial kit (Vazyme, Nanjing, China). The primers were synthesized by the Sangon Biotech Co., Ltd. (Shanghai, China). RT-qPCR was performed using the ChamQ SYBR qPCR Master Mix (Vazyme, Nanjing, China) in a Roche LightCycler 96 system (Roche, Basel, Switzerland). The PCR analysis was amplified as follows: 95 °C for 30 s, 40 cycles at 95 °C for 10 s, and 60 °C for 30 s. The analysis of the dissolution curve was as follows: 95 °C for 15 s, 60 °C for 60 s, and 95 °C for 15 s. GAPDH was used for normalization. The data were calculated using the 2^−^^△△Ct^ method [44]. The specific primer sequences for all the genes are listed in Table 4. Each sample was run in triplicate wells.

### 3.8. Chemical Assays of Antioxidant Activity

#### 3.8.1. DPPH Radical Scavenging Activity

The analytical method was modified from previously described methods [90]. Different MLF concentrations (1 mL) were mixed with 1 mL 0.004% DPPH/ethanol solution and 1 mL ethanol. Afterward, the mixtures were incubated in the dark for 30 min. The absorbance was measured at 517 nm. The DPPH radical scavenging activity was calculated using the following equation:DPPH scavenging activity (%) = [1 − (A_2_ − A_1_)/A_0_] × 100
where A_2_ is the absorbance value of the MLFs, A_1_ is the control group (95% ethanol instead of DPPH solution), and A_0_ is the blank group (distilled water instead of MLF).

#### 3.8.2. Metal Ion Chelating Activity Assay

The analytical method was modified from previously described methods [53]. Different MLF concentrations (1 mL) were mixed with FeCl_2_ (0.01 mL; 2 mmol/L) and ferrozine (0.04 mL; 5 mmol/L). The mixtures were allowed to stand for 10 min at 25 °C. The absorbance was measured at 562 nm. The chelating activity was calculated using the following equation:Chelating activity (%) = [1 − (A_2_ − A_1_)/A_0_] × 100
where A_0_ and A_2_ are the absorbance of the control (distilled water) and MLF solution, respectively, and A_1_ is the absorbance of the sample under identical A_2_ conditions (distilled water instead of FeCl_2_).

#### 3.8.3. Assessment of Reducing Power

The analytical method was modified from previously described methods [91]. Different MLF concentrations (2.5 mL) were combined with 2.5 mL phosphate-buffered saline (0.2 mol/L, pH = 6.6) and 2.5 mL 1% (*w*/*v*) K_3_Fe(CN)_6_ solution. Then, the mixture was incubated in a water bath at 50 °C for 20 min. Afterwards, 1 mL of 10% (*w*/*v*) TCA solution was added, and the mixture was centrifuged at 3000 rpm for 10 min. Then, 1 mL of the upper layer was mixed with 1 mL distilled water and 0.2 mL 0.1% (*w*/*v*) FeCl_3_ solution. The absorbance was measured at 700 nm. The reducing power was calculated using the following equation:Reducing power (%) = (A_1_/A_0_) × 100
where A_0_ and A_1_ are the absorbance of vitamin C and MLFs, respectively.

### 3.9. TFC Measurement

The TFC was measured following previously described methods with minor modifications [92]. Briefly, the MLFs were mixed with 1 mL 5% NaNO_2_ solution for 6 min, and then 1 mL of 10% Al (NO_3_)_3_ solution was added for 6 min. Subsequently, 10 mL of 4% NaOH solution was mixed with the reaction solution. The absorbance was measured at 510 nm using rutin as the standard.

### 3.10. MLF Sample Preparation and Liquid Chromatography-Mass Spectrometry (LC-MS) Conditions

The MLFs (0.05 g) were mixed with 10 mL of 80% methanol–water solution (*v*/*v*) in a 50-mL centrifuge tube and placed in a water bath at 70 °C for 30 min. Then, the samples were filtered through 0.22-μm membranes (Millipore, Billerica, MA, USA) into a glass bottle for the LC-MS analysis. Each sample had three replicates. Quality control (QC) samples were prepared by mixing 100 μL of three types of MLFs to evaluate the stability and reproducibility of the metabolomics analysis. Simply, it was connected to a quadrupole-time-of-flight (QTOF) MS (X500R; AB Sciex Co., Framingham, MA, USA) with an ultra-high-performance LC system (ExionLC, AB Sciex Co., Framingham, MA, USA). Samples were separated by a Zorbax Eclipse Plus C18 column (150 × 2.1 mm, 3.5 μm; Agilent Technologies, Little Falls, DE, USA). The mobile phase consisted of solvent A (water containing 0.1% formic acid, *v*/*v*) and solvent B (methanol) at a flow rate of 0.4 mL/min as follows: 10–15% B (0–4 min), 25–32% B (7–9 min), 40–55% B (16–22 min), 95% B (28–30 min), and 10% B (31–35 min). The injection volume was 10 μL. The parameters of electrospray ionization (ESI) were in positive ionization mode, and the Information Dependent Acquisition (IDA) method used ion source gases 1 and 2 at 50 psi. The CAD gas was set to 7 psi. The spray voltage, collision energy, and de-clustering voltage were 5500, 10, and 70 V, respectively. The mass scan range was 100–1000 Da.

### 3.11. Induction of Colitis and Treatment

Forty male 8-week-old C57BL/6J mice (20 ± 2 g) were purchased from Beijing HFK Bioscience Co., Ltd. (Beijing, China). All mice were freely provided with basic feed and distilled water. Before the animal experiments, all mice were acclimated for 7 days in an experimental animal laboratory. The mice were housed in a standard animal room at a controlled temperature (20 ± 5 °C) and constant humidity (40%–60%) under a 12-h light/12-h dark cycle. The experiment was approved by the Institutional Animal Care and Use Committee of the Northwest A&F University under permit number DK2022062.

In the experiments, 3% (*w*/*v*) DSS (36–50 kDa, MP Biomedicals, Santa Ana, CA, USA) was used to establish the mice UC model. All mice were randomly grouped as follows (*n* = 10): control group, DSS group, 30%MLF + DSS group, and 30%MLF group. The control and 30%MLF group mice were given distilled water, while the DSS and 30%MLF + DSS mice groups were given distilled water containing 3% DSS from the 4th to the 10th day for 7 days. The mice from the 30%MLF + DSS and 30%MLF groups were orally administered 500 mg/kg/d 30%MLF, while mice from the control and DSS groups received equal volumes of distilled water from the 1st day to the 10th day once a day. The experimental procedure is illustrated in Figure 11.

### 3.12. Assessment of DAI

The DAI assessment method was modified from previously described methods [93]. During the experiments, the mice were examined daily for body weight, stool consistency, and hematochezia in order to assess the colitis DAI. The standards for DAI were described in previous research [93]. The detailed method is shown in Appendix A.

### 3.13. Histological Analysis

The H&E method was modified from previously described methods [94]. A section of colon was cleaned with normal saline and fixed in 4% paraformaldehyde, followed by embedding, sectioning, and finally staining with H&E. The stained samples were observed and photographed using a microscope (Olympus, Tokyo, Japan). The criteria for the colon histopathological scores were based on a previous study with a slight modification [94]. The detailed method is displayed in Appendix A.

### 3.14. Immunohistochemical Staining

The method used for the immunohistochemical analysis was modified from previously described methods [95]. Briefly, the sectioned tissues were deparaffinized, rehydrated, and incubated with bovine serum albumin (BSA, 5%, Biosharp, Hefei, China) for 30 min. Then, the slices were incubated with specific primary antibodies for TLR4 (Bioss Antibodies, Beijing, China) and MyD88 (Boster Biological Technology Co. Ltd., Wuhan, China) at 4 °C overnight. After being washed three times with PBS, the sections were incubated with a secondary antibody (Servicebio Technology Co., Ltd. Wuhan, China) for 30 min. The slices were treated with diaminobenzidine (DAB; Servicebio Technology Co., Ltd., Wuhan, China), counterstained with hematoxylin, and visualized with a microscope (Olympus, Tokyo, Japan). For immunohistochemical staining, the average integrated positive area from three randomly selected regions was calculated using Image-Pro Plus 6.0 software (Media Cybernetics, Inc., Bethesda, MD, USA).

### 3.15. Data Treatment and Statistical Analysis

The LC-MS data were processed by using Markview software (AB Sciex, Framingham, MA, USA). The minimum and maximum retention times were set as 2 and 29 min, respectively. The retention time tolerance was set as 0.1 min, and the mass tolerance was set as 5 ppm.

A PCA and PLS-DA were performed using the Pareto scale (centered on the mean and divided by the square root of the standard deviation) using Simca-P v11.5 (Umetrics AB, Umeå, Sweden). A cluster analysis of the identified metabolites was performed after automatic scaling using MultiExperiment Viewer v4.9.0 (centered on the mean and divided by the square root of the standard deviation). The metabolites were clustered according to Pearson’s correlation.

The statistical analysis was conducted using SPSS v23.0 software (IBM Corporation, Armonk, NY, USA). All data were presented as the mean ± standard deviation. A Student’s *t*-test was used to determine the statistical significance. A *p*-value < 0.05 was considered statistically significant.

## 4. Conclusions

In the present study, the MLFs restrained NO and PGE2 production by decreasing the iNOS and COX-2 levels, as well as reduced the TNF-α, IL-1β, IL-6, and MCP-1 expression in LPS-induced RAW 264.7 macrophages. Compared to the LPS group, the fluorescence images showed that ROS were suppressed when treated with the MLFs. Moreover, the MLFs showed strong antioxidant properties based on the scavenging of DPPH free radicals, metal ion chelating activity, and the reducing power analysis. Among the MLFs, 30%MLF showed the greatest anti-inflammatory and antioxidant activities, the highest TFC, and had a high content of differential flavonoids, mainly containing quercetin, kaempferol, and their derivatives. Collectively, these results indicated that more flavonoids and the effective components of mulberry leaves could be obtained using a 30% alcohol concentration. In addition, the 30%MLF showed a protective effect in colitis mice induced by DSS. The 30%MLF treatment could mitigate the loss of body weight, decreased food intake, increased DAI, colon shorting, spleen edema, and colon damage in UC mice. More importantly, the 30%MLF treatment also effectively inhibited TNF-α, IL-1β, and IL-6 production and the activation of the TLR4/MyD88 pathway in UC mice. These results suggested that 30%MLF might be a potential agent to relieve UC.

## Figures and Tables

**Figure 1 ijms-23-07694-f001:**
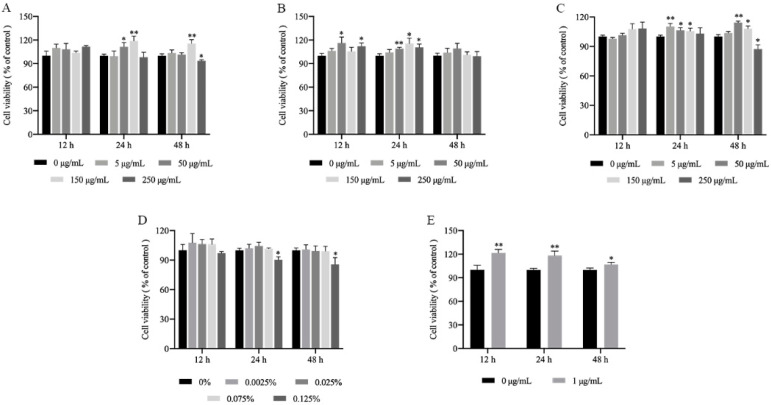
Cytotoxicity of (**A**) 30%MLF, (**B**) 50%MLF, (**C**) 75%MLF, (**D**) DMSO, and (**E**) LPS in RAW 264.7 cells for 12, 24, and 48 h. * *p* < 0.05 and ** *p* < 0.01 vs. 0 μg/mL.

**Figure 2 ijms-23-07694-f002:**
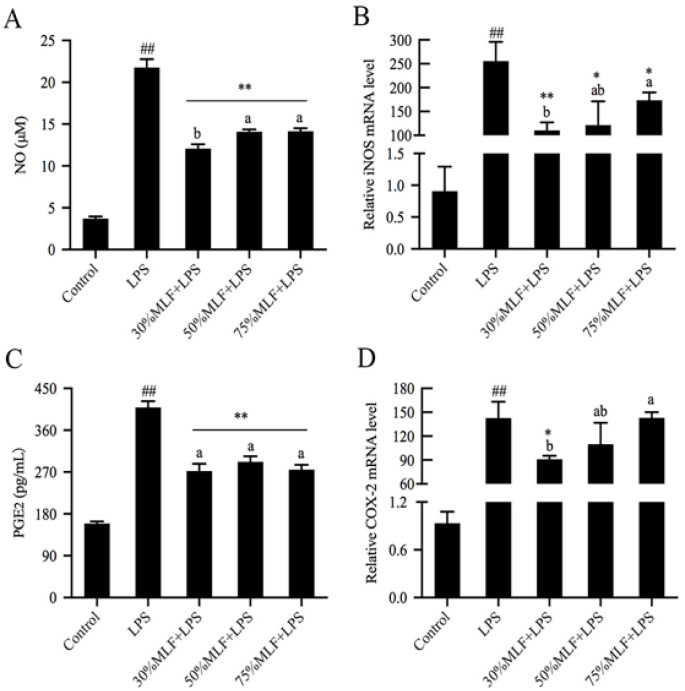
Effects of MLFs on (**A**) NO, (**B**) iNOS, (**C**) PGE2, and (**D**) COX-2 production in LPS-induced RAW 264.7 cells. Columns with different superscript letters are significantly different (*p* < 0.05); * *p* < 0.05 and ** *p* < 0.01 vs. the LPS group; ^##^
*p* < 0.01 vs. the control group.

**Figure 3 ijms-23-07694-f003:**
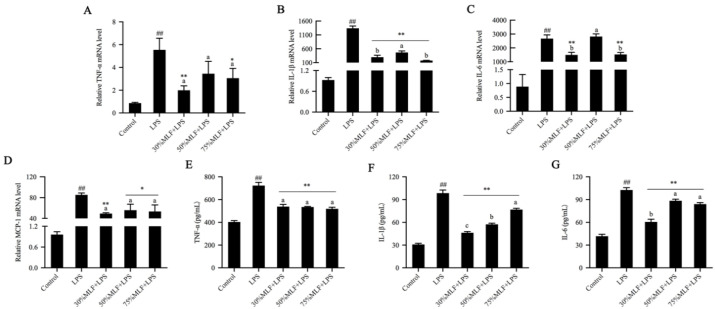
Effects of MLFs on inflammatory cytokine secretion in LPS-induced RAW 264.7 cells. The relative (**A**) TNF-α, (**B**) IL-1β, (**C**) IL-6, and (**D**) MCP-1 mRNA levels were quantitated by RT-qPCR; (**E**) TNF-α, (**F**) IL-1β, and (**G**) IL-6 production was measured by ELISA. Columns with different superscript letters are significantly different (*p* < 0.05). * *p* < 0.05 and ** *p* < 0.01 vs. the LPS group; ^##^
*p* < 0.01 vs. the control group.

**Figure 4 ijms-23-07694-f004:**
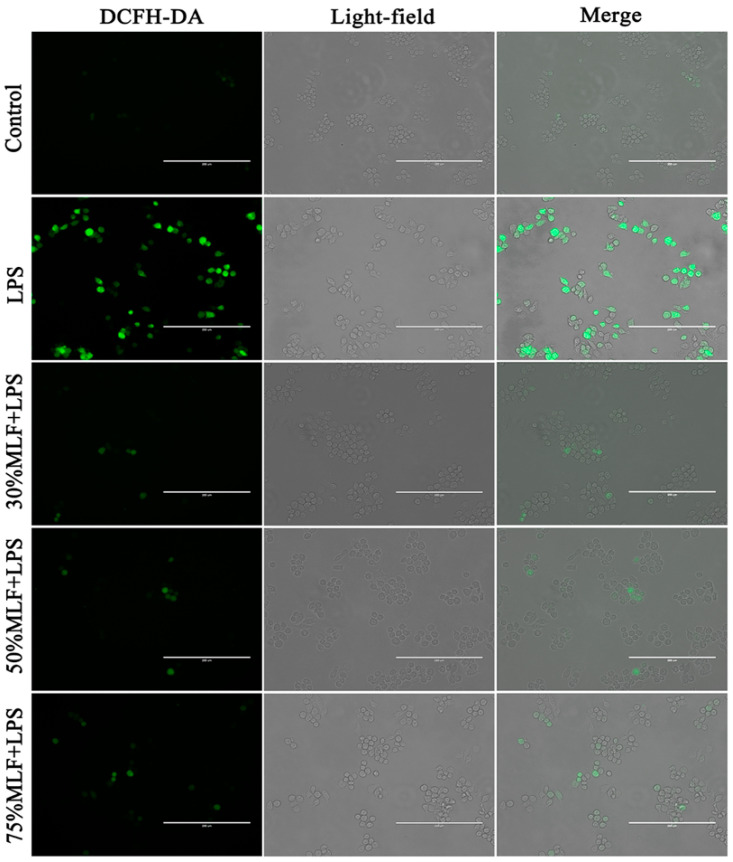
Effects of MLFs on ROS production in LPS-induced RAW 264.7 cells. DCFH-DA was used as a fluorescent probe to stain intracellular ROS, and fluorescence microscopy images show the intracellular generation of ROS. Scale bars, 200 µm.

**Figure 5 ijms-23-07694-f005:**
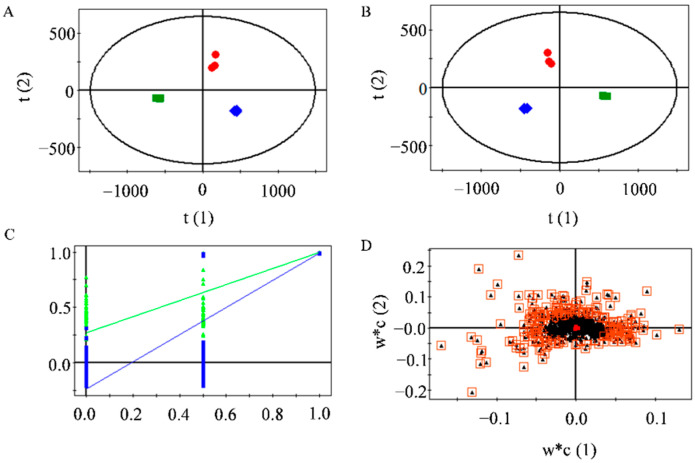
Multivariate statistical analysis of 30%MLF (squares), 50%MLF (circles), and 75%MLF (diamonds): (**A**) PCA score plot; (**B**) PLS-DA score plot, R^2^X = 0.91, R^2^Y = 0.99, Q^2^ = 0.979; (**C**) cross-validation plot of the PLS-DA model with 100 permutations, R^2^ = 0.273, Q^2^ = −0.236; and (**D**) PLS-DA loading plot—black triangles in boxes represent most differential metabolites.

**Figure 6 ijms-23-07694-f006:**
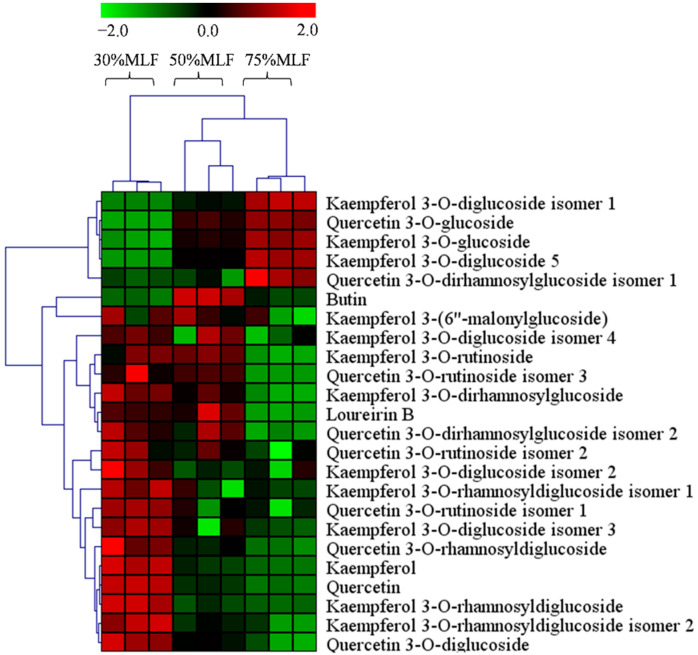
Heatmap of the flavonoid contents of 30%MLF, 50%MLF, and 75%MLF. Green indicates that the flavonoid level was less than the mean level in the MLFs, whereas red indicates that the flavonoid level was higher than the mean level.

**Figure 7 ijms-23-07694-f007:**
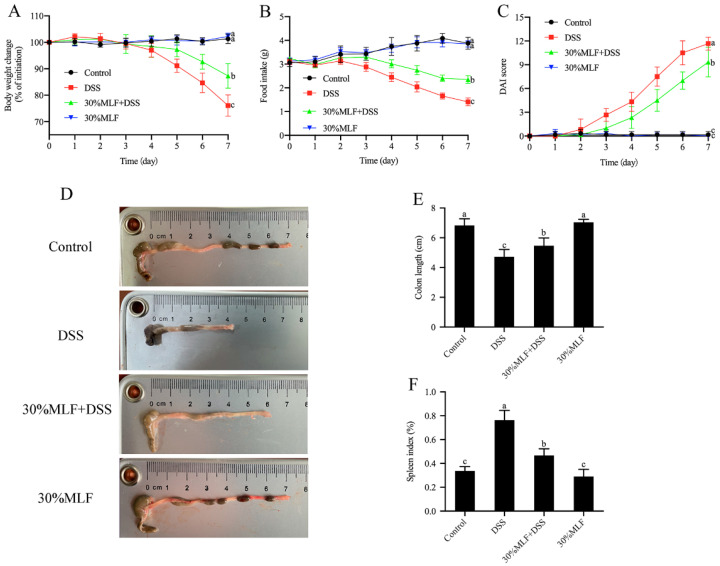
Effect of 30%MLF on the clinical symptoms of mice in DSS-induced ulcerative colitis: (**A**) body weight change and (**B**) food intake. (**C**) DAI. (**D**) Representative pictures of the macroscopic appearance of the colon length and (**E**) the statistical results. (**F**) Spleen index. Data are presented as the means ± SD (*n* = 6). Columns with different superscript letters indicate significant differences (*p* < 0.05).

**Figure 8 ijms-23-07694-f008:**
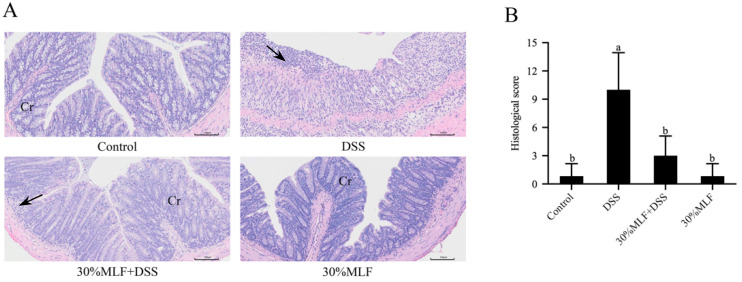
Effect of 30%MLF on the colonic pathological damage in mice with DSS-induced ulcerative colitis. (**A**) Representative images showing colon pathologic damages with H&E staining, where Cr indicates the crypt, and arrows indicate inflammatory cell infiltration. (**B**) Histological scores of colons. Data are presented as the means ± SD (*n* = 6). Columns with different superscript letters are significantly different (*p* < 0.05).

**Figure 9 ijms-23-07694-f009:**
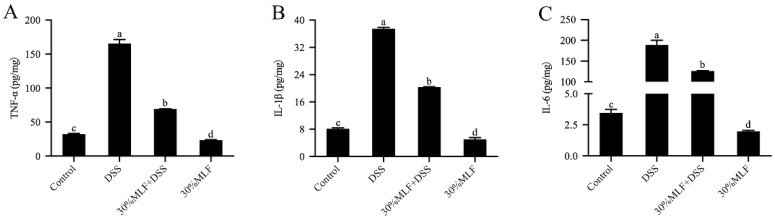
Effects of 30%MLF on the inflammatory cytokine secretion of mice with DSS-induced ulcerative colitis. (**A**) TNF-α, (**B**) IL-1β, and (**C**) IL-6 production in colon tissue was measured by ELISA. Data are presented as the means ± SD (*n* = 3). Columns with different superscript letters are significantly different (*p* < 0.05).

**Figure 10 ijms-23-07694-f010:**
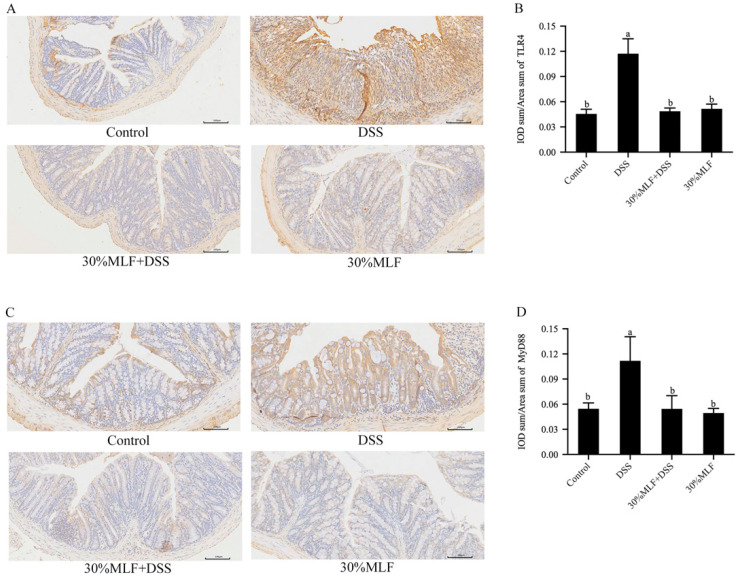
Effects of 30%MLF on the immunohistochemical analysis of (**A**,**B**) TLR4 and (**C**,**D**) MyD88 in the colon tissue of mice in DSS-induced UC. The average positive area from three randomly selected regions was calculated using Image-Pro Plus 6.0 software. Data are presented as the means ± SD (*n* = 3). Columns with different superscript letters are significantly different (*p* < 0.05).

**Figure 11 ijms-23-07694-f011:**
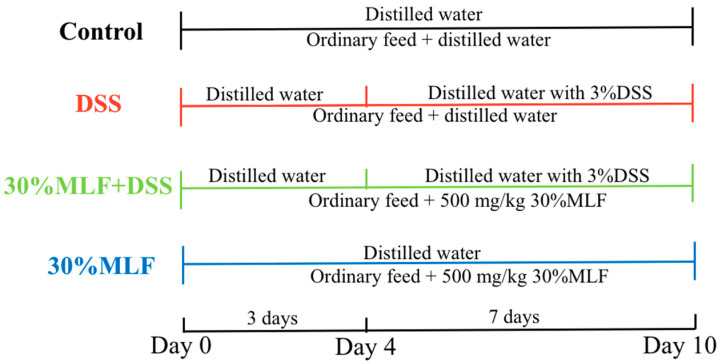
Experimental design of DSS-induced colitis in mice.

**Table 1 ijms-23-07694-t001:** Antioxidative activities of the MLFs.

MLF	IC_50_^1^ (μg/mL)		Reducing Power (%)
	DPPH Scavenge	Chelating Activity	
30%MLF	12.57 ± 0.90 ^b^	310.56 ± 9.72 ^c^	41.69 ± 2.41 ^b^
50%MLF	12.35 ± 0.55 ^b^	389.83 ± 14.14 ^b^	38.95 ± 0.45 ^b^
75%MLF	19.53 ± 1.16 ^a^	482.19 ± 34.74 ^a^	31.11 ± 0.39 ^c^
Vitamin C	6.01 ± 0.50 ^c^		100.00 ± 0.06 ^a^
EDTA		1.72 ± 0.27 ^d^	

Each value shows the mean ± SD (*n* = 3). Means with different lowercase letters within the same column denote significant differences (*p* < 0.05). Positive controls: ethylenediaminetetraacetic acid (EDTA) and vitamin C. IC_50_^1^ represents the concentration of the MLFs scavenging 50% of the free radical and chelating 50% of Fe^2+^.

**Table 2 ijms-23-07694-t002:** Analysis of the total flavonoid content (TFC) of the MLFs.

MLF	TFC (%)
30%MLF	72.89 ± 0.18 ^a^
50%MLF	61.84 ± 0.04 ^b^
75%MLF	55.12 ± 0.50 ^c^

Each value shows the mean ± SD (*n* = 3). Means with different lowercase letters within the same column denote significant differences (*p* < 0.05).

**Table 3 ijms-23-07694-t003:** Different flavonoids putatively identified among the MLFs.

Compound Name	Retention Time/min	Detected Mass (ESI+)	Ion Types	MS/MS Fragments
Butin	21.76	273.0755	M + H	137, 81
Loureirin B	20.88	317.1382	M + H	299, 167, 149, 121
Kaempferol ^a^	22.32	287.0548	M + H	287, 153, 135, 107
Kaempferol 3-(6’’-malonylglucoside)	22.88	535.1085	M + H	287
Kaempferol 3-*O*-diglucoside isomer 1	10.27	611.1606	M + H	449, 287
Kaempferol 3-*O*-diglucoside isomer 2	10.84	611.1616	M + H	449, 287
Kaempferol 3-*O*-diglucoside isomer 3	8.68	611.1622	M + H	449, 287
Kaempferol 3-*O*-diglucoside isomer 4	13.14	611.1626	M + H	449, 287
Kaempferol 3-*O*-diglucoside 5	15.42	633.1445	M + Na	633, 347
Kaempferol 3-*O*-dirhamnosylglucoside	17.66	741.2248	M + H	595, 449, 287
Kaempferol 3-*O*-rhamnosyldiglucoside	12.3	795.1751	M + H	644
Kaempferol 3-*O*-rhamnosyldiglucoside isomer 1	8.644	757.2204	M + H	611, 449, 287
Kaempferol 3-*O*-rhamnosyldiglucoside isomer 2	10.58	757.2208	M + H	611, 449, 287
Kaempferol 3-*O*-rutinoside 1	21.82	595.1658	M + H	449, 287
Kaempferol-3-*O*-glucoside ^a^	17.47	449.1077	M + H	287
Quercetin ^a^	14.65	303.0496	M + H	257, 229, 165, 153, 137
Quercetin 3-*O*-dirhamnosylglucoside isomer 1	12.28	757.219	M + H	611, 465, 303
Quercetin 3-*O*-dirhamnosylglucoside isomer 2	11.76	757.2207	M + H	611, 465, 303
Quercetin 3-*O*-rhamnosyldiglucoside 3	9.07	773.2145	M + H	627, 465, 303
Quercetin-3-*O*-rutinoside isomer 1	12.57	611.1608	M + H	611, 465, 303
Quercetin-3-*O*-rutinoside isomer 2	14.99	611.1614	M + H	465, 303
Quercetin-3-*O*-rutinoside isomer 3	11.73	611.1621	M + H	465, 303
Quercetin-diglucoside 4	8.86	627.1575	M + H	465, 303
Quercetin-3-*O*-glucoside ^a^	10.84	465.1026	M + H	303, 153, 149

^a^ Confirmed by the authentic standards.

**Table 4 ijms-23-07694-t004:** Primer sequences used in RT-qPCR.

Genes		Primer Sequences (from 5′ to 3′)
TNF-α	Forward	CCACGCTCTTCTGTCTACTG
	Reverse	ACTTGGTGGTTTGCTACGAC
IL-1β	Forward	CCAACAAGTGATATTCTCCATGAG
	Reverse	ACTCTGCAGACTCAAACTCCA
IL-6	Forward	CTCTGCAAGAGACTTCCATCC
	Reverse	GAATTGCCATTGCACAACTC
iNOS	Forward	TTTCCAGAAGCAGAATGTGACC
	Reverse	AACACCACTTTCACCAAGACTC
COX-2	Forward	GAAATATCAGGTCATTGGTGGAG
	Reverse	GTTTGGAATAGTTGCTCATCAC
MCP-1	Forward	AAGAAGCTGTAGTTTTTGTCACCA
	Reverse	TGAAGACCTTAGGGCAGATGC
HO-1	Forward	ACATTGAGCTGTTTGAGGAG
	Reverse	TACATGGCATAAATTCCCACTG
GAPDH	Forward	GAGAAACCTGCCAAGTATGATGAC
	Reverse	TAGCCGTATTCATTGTCATACCAG

## Data Availability

The data presented in this study are available in the article and the Appendix A.

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
