# Peer review of "Anti-Inflammatory Activity of Mulberry Leaf Flavonoids In Vitro and In Vivo"

_ijms, 2022, doi:10.3390/ijms23147694_

Round 1
Reviewer 1 Report
The resubmitted version of the manuscript has been sufficiently improved.
Author Response
Thank you for your recognition. We are very glad to hear this comment.
Reviewer 2 Report
The current resubmitted manuscript by Lin et al. is a significantly improved version of the original manuscript and may be considered for publication in International Journal of Molecular Sciences pending the following major revisions.
1. The authors should duplicate their cell line (in vitro) studies in at least one more macrophage cell line.
2. The authors should discuss their findings in Figure 1 in more detail. The authors should clarify and explain why both MLFs that exhibit anti-inflammatory activity and LPS that exhibits proinflammatory activity show an increase in the viability of the RAW264.7 cells.
Author Response
Please see the attachment

This manuscript is a resubmission of an earlier submission. The following is a list of the peer review reports and author responses from that submission.
Round 1
Reviewer 1 Report
The current manuscript by Lin et al. though interesting in its findings requires additional experiments for attaining the scientific soundness and significance of the content to warrant publication in the International Journal of Molecular Sciences. In light of this, I recommend to reject the publication of this manuscript in its current form in the International Journal of Molecular Sciences.
The authors should perform the additional experiments as below and consider resubmission to International Journal of Molecular Sciences.
1. The authors should perform the studies in at least one more macrophage cell line.
2. The authors should perform in vivo studies.
In addition, the authors should address some minor concerns as noted below:
1. The authors should determine and provide the effect of LPS induction on macrophage cell viability.
2. In the results & discussion section, while describing the findings shown in Fig. 1, the authors claim that 50% MLF was not toxic and that 30% and 75% MLF significantly affected cell viability. The data shown in Fig. 1 reveals that 30%, 50%, and 75% MLF are all not toxic and affect cell viability significantly either at 12 or 24 or 48 hours. The authors should clarify this and describe their findings more clearly.
3. In line# 92-93, the authors say that "MLF at 250 μg/mL significantly affected cell viability after 48 h" but go ahead to say in line# 93-94 that "150 μg/mL was selected for use in the experiments". The authors should clarify which dosage, 250 or 150 μg/mL was used in the further experiments.
4. In line# 94-95, the authors say that "0.0075% DMSO had no obvious toxic effect on the macrophages (Fig. 1D)" but Fig. 1D does not show a DMSO dosage of 0.0075% being used for this study. The authors should clarify this and describe their findings more clearly.
5. In fig. 2 & 3 rather than using different lower case letters to show statistical significance in LPS treated group with MLFs, the authors should provide p values for comparison among the various MLFs treated groups (what was the p value for 30% vs 50% MLF, 30% vs 75% MLF, and 50% vs 75% MLF).
Reviewer 2 Report
After reviewing the manuscript, I found critical concerns that need to be clarified and addressed. Therefore, I recommend the authors consider the following points.
Major points:
Introduction section
- In the Introduction section, I recommend the authors mention that other parts of Morus alba such as root bark are rich in flavonoids with anti-inflammatory properties. This information can be extracted from the recommended reference (DOI: 10.1016/j.jep.2019.112296).
Materials and Methods section
- There are no statistical analyses described in this section. The authors should provide detailed information about the used statistical analyses for all performed experiments (statistical tests; for example, what tests were used to determine the differences between treatments with test samples and the positive controls, along with information about post-hoc comparison tests and statistical significance). The corresponding results cannot be accepted without a statistical analysis validation.
- Please provide information about the plant and its authorized identification along with the amount of the used samples.
- All experiments should be performed according to proper methodologies. Please add the proper citation for each experiment.
- Where are the experiments for determining the total flavonoid content (TFC) for the three MLFs extracts? This information should be added.
- In the Materials and Methods section, all experiments that are mentioned in the Results and Discussion section should be experimentally described in detail. Several methods are missed.
- Most importantly is that the authors should provide chromatographic data (HPLC-MS) of all identified flavonoid compounds in all analyzed MLFs extracts. Also, the used method should be described in detail along with the chromatographic, operational, and instrumental setting parameters. Another important point, did the authors use standard flavonoid compounds for identification and prepared calibration curves for identified compounds? Also, in my opinion, more compounds should be presented in all analyzed MLFs extracts. So, if the authors would provide chromatograms of HPLC-MS, it would be clear how many compounds are presented in all analyzed MLFs extracts. Additionally, the authors should describe how they set the instrumental parameters to separate and identify only flavonoid compounds and avoid the separation of other compounds such as terpenoids, alkaloids, and other substances. This is a very important point and should be addressed and clarified.
Minor points:
- I recommend the authors double-check the full text for grammatical and typing errors.
Round 2
Reviewer 1 Report
The authors have addressed the minor concerns but are unwilling to address the two major concerns stated below that was noted in previous review.
1. The authors should perform the studies in at least one more macrophage cell line
2. The authors should perform in vivo studies.
In light of this, the authors' response is unsatisfactory and I recommend to reject the manuscript for publication in International Journal of Molecular Sciences.
Reviewer 2 Report
The manuscript has been sufficiently improvrd.